Effect of heat-killed Streptococcus thermophilus on type 2 diabetes rats

Gao Xiangyang 1
Wang Fei 1
Zhao Peng 1 2
Zhang Rong 1
Zeng Qiang 1 zengqianghospital@126.com
1 Health Management Institute, The Second Medical Center of Chinese PLA General Hospital , Beijing , China
2 Health Management Center, HangZhou Special Service Convalescent Center of Air Force, PLA , Hangzhou , China
Foti Daniela
Electronic publication date: 2019 Jun 13
Publication date: 2019
Volume: 7
Electronic Location ID: e7117
Received 2019 Feb 4; Accepted 2019 May 10
Copyright: © 2019 Gao et al.
Copyright year: 2019
Copyright holder: Gao et al.
License: This is an open access article distributed under the terms of the Creative Commons Attribution License, which permits unrestricted use, distribution, reproduction and adaptation in any medium and for any purpose provided that it is properly attributed. For attribution, the original author(s), title, publication source (PeerJ) and either DOI or URL of the article must be cited.
License URL: https://creativecommons.org/licenses/by/4.0/

Keywords: Diabetes, Gut microbiota, Heat-killed Streptococcus thermophilus, ZDF rats

Funding: Australia-China International Collaborative Grant NH&MRC-APP1112767-NSFC81561128020-0023 National Natural Science Foundation of China 81872920 This work was supported by Australia-China International Collaborative Grant (NH&MRC-APP1112767-NSFC81561128020-0023) and National Natural Science Foundation of China (81872920). The funders had no role in study design, data collection and analysis, decision to publish, or preparation of the manuscript.

==============================
Background and Aims

The link between gut microbiota and type 2 diabetes (T2D) has been addressed by numerous studies. Streptococcus thermophilus from fermented milk products, has been used as a probiotic in previous research. However, whether heat-killed S. thermophilus can improve the glycemic parameters of diabetic rats remains unanswered. In this study, we evaluated the effect of heat-killed S. thermophilus on T2D model rats and the potential mechanisms of the effect.

Methods

Zucker diabetic fatty (ZDF) rats were used to generate a diabetic rat model induced by feeding a high-fat diet. Heat-killed S. thermophilus were orally administered to normal and diabetic rats for 12 weeks. Intestinal microbiota analysis, histology analysis, oral glucose tolerance test and measurement of inflammatory factors were performed.

Results

We found that heat-killed S. thermophilus treatment reduced fasting blood glucose levels and alleviated glucose intolerance and total cholesterol in diabetic ZDF rats. Additionally, heat-killed S. thermophilus increased the interleukin 10 while reducing the levels of lipopolysaccharide, interleukin 6, and tumor necrosis factor-α in diabetic ZDF rats. The heat-killed S. thermophilus treatment can normalize the structure of the intestinal and colon mucosal layer of diabetic rats. The characteristics of the gut microbiota in heat-killed S. thermophilus-treated and control rats were similar. At the genus level, the abundances of beneficial bacteria, including Ruminococcaceae, Veillonella, Coprococcus, and Bamesiella, were all significantly elevated by heat-killed S. thermophilus treatment in ZDF diabetic rats.

Conclusion

Our study supports the hypothesis that treatment with heat-killed S. thermophilus could effectively improve glycemic parameters in T2D model rats. In addition, the potential mechanisms underlying the protection maybe include changing the composition of gut microbiota, reinforcing the intestinal epithelial barrier and the immunity of the intestinal mucosa, decreasing the level of inflammation, and then reducing the insulin resistance.

Introduction

Diabetes is a chronic metabolic disease and an important cause of mortality and morbidity worldwide, the prevalence of which is dramatically increasing. The number of adults with diabetes, mostly Type 2 diabetes (T2D), has increased to 422 million around the world (NCD Risk Factor Collaboration (NCD-RisC), 2016). Diabetes and its complications account for more than two million deaths every year (Li et al., 2016).

Recently, immense evidence has been obtained linking T2D and gut microbiota. The significant correlations with specific gut microbes, bacterial genes, and metabolic pathways in T2D patients were showed by a human metagenome-wide association study (Larsen et al., 2010). The data from animal and human models also suggest that T2D is associated with a moderate degree (Qin et al., 2012) to profound gut microbial dysbiosis (Tilg & Moschen, 2014). Increasing evidence indicates that gut microbiota are strongly associated with diabetes development (Haro et al., 2015; Mejía-León & Barca, 2015). Other studies even show that gut microbiota markedly contribute to the incidence of T2D (Baothman et al., 2016; Tai, Wong & Wen, 2015). The dysbiosis of gut microbiota may damage the intestinal epithelial barrier, and increase the intestinal permeability, and thus promotes metabolic endotoxemia, and systemic inflammation (Prattichizzo et al., 2018; Winer et al., 2016), leading to the development of insulin resistance (Jorge et al., 2012; Boulange et al., 2016), thereby increasing the risk of developing T2D (Pedersen et al., 2016; Hartstra, Nieuwdorp & Herrema, 2016). These studies suggest that the gut microbiota are potential targets for the treatment of T2D.

Probiotics have been proven to be effective in T2D. Administration of probiotics in a rat model effectively inhibited gluconeogenesis in T2D (Amandine et al., 2013). Treatments with probiotics have been demonstrated to be efficacious against tissue inflammation, and insulin resistance by modulating the gut microbial structure (Moya-Pérez, Neef & Sanz, 2015; Shin et al., 2014). However, the efficacy in T2D subjects varies, depending on the types and strains of probiotics.

Probiotics, as defined by the World Health Organization, are live microorganisms, that confer a health benefit to the host, when administered in adequate amounts (FAO/WHO, 2001). However, in many cases, probiotic preparations comprised of dead cells and their metabolites can also exert a biological response similar to that seen with live cells (Dotan & Rachmilewitz, 2005; Sashihara, Sueki & Ikegami, 2006; Zhang et al., 2005). For example, both live and heat-killed Lactobacillus GG had a similar anti-inflammatory effect (Ehud et al., 2004).

Streptococcus thermophilus is classified as a lactic acid bacterium, and it is found in fermented milk products, and generally used in the dairy industry (Kilic et al., 1996). S. thermophilus scavenges reactive oxygen radicals (Lin & Yen, 1999; Bruno-Barcena et al., 2004), thus demonstrating its antioxidant properties. S. thermophilus also shows immunomodulatory effects by stimulating the gut immune system (Donkor et al., 2012; Delorme, 2008). And S. thermophilus has been used as a probiotic to help prevent developing insulin resistance in previous research (Asemi et al., 2013a). However, to our knowledge, the question as to whether heat-killed S. thermophilus can improve glycemic parameters remains unanswered. In addition, the potential mechanisms underlying the possible protection are still poorly understood. Therefore, the purpose of this research was to identify the beneficial effects of heat-killed S. thermophilus on diabetic rats and the potential mechanisms.

Materials and Methods

T2D animal model

The Zucker diabetic fatty (ZDF) rats were used as a T2D model. ZDF rats have been an important model for studying the mechanism of treatment on T2D (Finegood et al., 2001; Leonard et al., 2005). Seven-week-old male ZDF rats were purchased from Charles River (Beijing, China). After 1 week of acclimation, diabetes was then induced by feeding a high-fat diet of Purina5008 (17% kcal fat and 26.5% kcal protein; IPS Supplies, London, UK) for 1 month. Then, 12-week-old male ZDF rats were obtained, and fasting blood glucose (FBG) >11.1 mmol/l was determined to be the standard concentration for the T2D model.

Control rats

Seven-week-old male Sprague-Dawley (SD) rats also were obtained from Charles River (Beijing, China). After acclimating for 1 week, they were used as control rats.

Both the ZDF and SD rats were maintained at 22 ± 2 °C with lights in an air-conditioned room with a 12-h light/dark cycle, and were given free access to food and water. A standardized diet (kcal%: 10% fat, 20% protein, and 70% carbohydrate) was administered. All of the experimental protocols were approved by the Animal Care Committee of the General PLA Hospital Animal Ethics Committee (Project CPLAGHAE-20171228-01).

Study design

The diabetic ZDF rats were randomly divided into two groups: a heat-killed S. thermophilus-treated diabetic group (DM+ST, KAWAI; Kawai Lactic Acid Bacteria Research Institute Co., Ltd., Tokyo, Japan, orally administered 0.21 g Kawai powder/kg body weight/day, n = 5) and an untreated diabetic group (DM, orally administered the same volume of normal saline, n = 5). Kawai powder contains 28.75% heat-killed S. thermophilus and 20.60% resistant dextrin, 20.00% isomaltooligosaccharide, 17.00% microcrystalline cellulose, 10.00% xylo-oligosaccharides, 2.55% Saccharomyces cerevisiae, and 1.10% lemon juice powder.

Control rats were randomly divided into an untreated control group (CON, administered normal saline, n = 5) and a heat-killed S. thermophilus-treated control group (CON+ST, orally administered 0.21 g Kawai powder/kg body weight/day, n = 5). After treatment for 12 weeks, fresh stool samples were obtained by stimulating the anus, and they were frozen and stored at −80 °C for subsequent analysis. After food deprivation for 12 h, the rats were anesthetized, blood samples were collected from the aorta abdominalis, and then the rats were sacrificed.

Tissue collection and histology analysis

After rats were killed, the tissues of the ileum and colon were immediately excised, and then were cleaned with ice-cold phosphate-buffered saline solution. The tissues were fixed in 4% formalin solution, then embedded in paraffin before being cut into four-μm slices, followed by hematoxylin-eosin staining for measurement of villi length and crypt depth (10 villi and 10 crypts per section) under a light microscope (SZX16; Olympus, Tokyo, Japan).

Western blot analysis

The ileum and colon tissues were homogenized in RIPA lysis buffer containing protease inhibitor cocktail (Roche, Indianapolis, IN, USA). Protein homogenates were separated on SDS-PAGE gels and transferred to polyvinylidene difluoride membranes. After blocking for 1 h with 5% bovine serum albumin in Tris-buffered saline with 0.1% Tween (TBST: 50 mM Tris–HCl, 150 mM NaCl, 0.1% Tween 20, pH 7.4), the membranes were incubated overnight with specific primary antibodies against Occludin (Abcam, Cambridge, UK), ZO-1 (Zonula occludens) (Santa Cruz Biotechnology, Dallas, TX, USA), and β-actin (Zsbio, Beijing, China) at 4 °C. Then, the membranes were incubated for 1 h with the appropriate horseradish peroxidase (HRP)-conjugated secondary antibodies (anti-rabbit or anti-mouse IgG-HRP) (Jackson ImmunoResearch Inc., West Grove, PA, USA), and the bands were detected by using enhanced chemiluminescence. The blots were scanned by a Bio-Rad ChemiDoc XRS and the intensity of each protein was quantified by Gel Image system V4.00 software (Tanon, Shanghai, China).

Oral glucose tolerance test

At the end of the trial, an oral glucose tolerance test (OGTT) was performed after fasting for 12 h. Glucose (two g/kg body weight) was orally administered to the rats. The blood glucose levels which were obtained from the tail were recorded with a OneTouch UltraEasy glucometer (Johnson & Johnson, New Brunswick, NJ, USA) before and 15, 30, 60, 90, and 120 min after the glucose load. The area under the curve (AUC) was calculated by using the linear trapezoid method (Zhang et al., 2016).

Measurement of inflammatory factors, serum insulin, lipid profile, HOMA-IR, and HbA1c

After food deprivation for 12 h, rat serum was obtained to analyze inflammatory factors (interleukin 6 (IL-6), interleukin 10 (IL-10), tumor necrosis factor (TNF)-α, and lipopolysaccharide (LPS) (ELISA, Elabscience, Wuhan, China), insulin (ELISA, Millipore, Billerica, MA, USA), total cholesterol (TC), triglyceride (TG, oxidase method; InTec Products, Fujian, China), high-density lipoprotein cholesterol concentrations (HDL-C), and low-density lipoprotein cholesterol concentrations (LDL-C, direct method, InTec Products, Fujian, China), according to the manufacturer’s instruction. The homeostasis model assessment of insulin resistance (HOMA-IR) was calculated by using the following formula: FBG (mmol/l) × fasting serum insulin (μIU/ml)/22.5. Rat plasma was also analyzed for HbA1c (Immunoturbidimetry; InTec Products, Fujian, China).

Intestinal microbiota analysis

DNA extractions from total fecal bacteria were obtained using a QIAamp Stool DNA Extraction Kit (Qiagen, Valencia, CA, USA) according to the manufacturer’s instructions. The microbial 16S rRNA hypervariable regions V3–V4 were amplified with indexes and adaptor-linked universal primers (341F: 50-ACTCCTACGGGAGGCAGCAG-30, 806R: 50-GGACTACHVGGGTWTCTAA-30T). PCR was performed by using a KAPA HiFi Hotstart PCR kit (KAPA Biosystems, Wilmington, DE, USA) with high fidelity enzyme in triplicate. Amplicon libraries were quantified using a Qubit 2.0 Fluorometer (Thermo Fisher Scientific, Waltham, MA, USA) and then sequenced on the Illumina HiSeq 2500 platform (Illumina, San Diego, CA, USA) for 250-bp paired-end reads. After discarding the singletons and removing chimeras, operational taxonomic units (OTUs) were generated using USEARCH (v7.0.1090) at 97% similarity by clustering the tags. Final OTUs were taxonomically classified based on the RDP classifier version 2.2 algorithm using the GreenGene database. Alpha diversity (Chao1, Shannon, Simpson) and beta diversity (principal coordinates analysis (PCoA) plots) were analyzed using QIIME version 1.7.0. In addition, a t-test was performed to compare the differences between groups by using STAMP. The relative abundance of bacteria is expressed as the percentage (%).

Data analysis

The data are expressed as the mean ± standard deviation (SD). When the data were normal and variances were equal, differences among the groups were analyzed using t-test. For non-normal distribution data, ln transformation was carried out before analysis. A p-value < 0.05 was considered statistically significant. All of the statistical analyses were performed using the Statistical Package for Social Sciences version 17 software (SPSS Inc., Chicago, IL, USA).

Results

Body weight

The body weights of the heat-killed S. thermophilus–treated diabetic rats were comparable with those of the untreated diabetic rats (p > 0.05, Table 1). There was also no significant difference in body weight between the CON group and CON+ST group (p > 0.05, Table 2).

Table 1 The differences in some variables between the DM+ST and DM groups.

Variables	DM	DM+ST	t	p	
Weight (g)	354.2 ± 35.2	360.2 ± 33.0	−0.3	0.788	
Creatinine (μmol/l)	16.0 ± 6.2	15.0 ± 2.1	0.3	0.742	
ALT (U/l)	119.4 ± 51.5	123.4 ± 52.7	−0.1	0.906	
Carbamide (mmol/l)	6.0 ± 0.8	7.2 ± 1.6	−1.6	0.159	
Uric acid (umol/l)	116.2 ± 32.6	122.6 ± 18.2	−0.4	0.711	
TC (mmol/l)	5.5 ± 0.4	4.7 ± 0.2	4.1	0.003*	
Triglyceride (mmol/l)	3.0 ± 0.9	2.5 ± 0.3	1.2	0.304	
HDL-C (mmol/l)	2.7 ± 0.2	2.5 ± 0.2	1.8	0.115	
LDL-C (mmol/l)	1.2 ± 0.2	0.9 ± 0.1	2.5	0.063	
LPS (ng/ml)	0.7 ± 0.1	0.5 ± 0.1	2.9	0.019*	
LNIL6 (pg/ml)	4.6 ± 0.6	3.9 ± 0.3	2.7	0.038*	
LNIL10 (pg/ml)	4.0 ± 0.3	4.5 ± 0.3	−2.4	0.046*	
LnTNF-α (pg/ml)	4.1 ± 0.3	3.6 ± 0.2	3.0	0.017*	
Fasting insulin (μIU/ml)	107.6 ± 18.1	67.0 ± 8.3	4.6	0.002*	
HOMA-IR	106.7 ± 25.5	40.8 ± 3.8	5.7	0.004*	
HbA1c (%)	12.0 ± 2.0	8.6 ± 1.2	3.3	0.011*	
FBG (mmol/l)	22.3 ± 3.6	13.7 ± 1.0	3.1	0.036*	
Notes:

N = 5 in each group. Data represented as means ± SD. The heat-killed S. thermophilus treatment reduced TC, LPS, IL-6, IL-10, TNF-α, fasting insulin levels, HbA1c, FBG, and HOMA-IR in ZDF diabetic rats.

ALT, alanine aminotransferase; TC, total cholesterol; HDL-C, high-density lipoprotein cholesterol; LDL-C, low-density lipoprotein cholesterol; LNIL6, ln transformation of interleukin-6; LNIL10, ln transformation of interleukin-10; LnTNF-α, ln transformation of tumor necrosis factor-α; LPS, lipopolysaccharide; HOMA-IR, homeostasis model assessment of insulin resistance; FBG, fasting blood glucose.

* p < 0.05.

Table 2 The differences in some variables between the CON+ST and CON groups.

Variables	CON+ST	CON	t	p	
Weight (g)	503.2 ± 48.5	476.8 ± 37.3	1.0	0.363	
Creatinine (μmol/l)	24.2 ± 2.8	26.2 ± 4.7	−0.8	0.437	
ALT (U/l)	37.6 ± 9.8	36.8 ± 7.6	0.1	0.889	
Carbamide (mmol/l)	6.1 ± 1.4	5.7 ± 1.5	0.4	0.709	
Uric acid (μmol/l)	122.8 ± 21.7	154.0 ± 66.6	−1.0	0.348	
TC (mmol/l)	1.4 ± 0.3	1.8 ± 0.4	−1.8	0.11	
Triglyceride (mmol/l)	0.7 ± 0.2	0.7 ± 0.3	0.1	0.963	
HDL-C (mmol/l)	0.8 ± 0.2	0.9 ± 0.2	−1.3	0.246	
LDL-C (mmol/l)	0.3 ± 0.1	0.4 ± 0.1	−1.6	0.16	
LPS (ng/ml)	0.4 ± 0.1	0.4 ± 0.1	−1.0	0.369	
lnln6 (pg/ml)	4.0 ± 0.2	4.1 ± 0.3	−0.6	0.585	
lnlL10 (pg/ml)	4.3 ± 0.5	4.1 ± 0.1	0.8	0.465	
lnTNF-α (pg/ml)	3.9 ± 0.4	4.1 ± 0.3	−1.1	0.323	
Fasting insulin (μIU/ml)	41.1 ± 7.0	42.2 ± 12.5	−0.2	0.863	
FBG (mmol/l)	6.7 ± 0.6	6.9 ± 0.6	−0.4	0.72	
HOMA-IR	12.3 ± 2.8	12.7 ± 2.9	−0.2	0.857	
Notes:

N = 5 in each group. Data represented as means ± SD. Compared to the CON group, the heat-killed S. thermophilus treatment rats failed to show significant variations.

ALT, alanine aminotransferase; TC, total cholesterol; HDL-C, high-density lipoprotein cholesterol; LDL-C, low-density lipoprotein cholesterol; LNIL6, ln transformation of interleukin-6; LNIL10, ln transformation of interleukin-10; LnTNF-α, ln transformation of tumor necrosis factor-α; LPS, lipopolysaccharide; HOMA-IR, homeostasis model assessment of insulin resistance; FBG, fasting blood glucose.

Fasting blood glucose level and glucose tolerance

The heat-killed S. thermophilus treatment reduced FBG levels in diabetic rats (p < 0.05, Fig. 1). The blood glucose levels significantly decreased before and 15, 60, and 90 min after glucose load (p < 0.05, Fig. 1) in the DM+ST group as compared to those in the DM group according to the OGTT. At the time points of 30 and 120 min after the glucose load, the blood glucose levels were lower in the DM+ST group than those in DM group, but the differences were not significant (p > 0.05, Fig. 1). Compared with the DM group, the glucose AUC for the OGTT in the DM+ST group exhibited a reduced glucose AUC by 14.7% (p < 0.05, Fig. 2).

Figure 1 The effect of heat-killed S. thermophilus on blood glucose during the OGTT in the DM+ST and DM groups.

OGTT, oral glucose tolerance test. Error bars represent one standard deviation. The blood glucose levels significantly decreased before and 15, 60, and 90 min after glucose load in the DM+ST group as compared to those in the DM group according to the OGTT. *p < 0.05.

Figure 2 Area under the curve (AUC) for the OGTT in the DM+ST and DM groups.

Compared with the DM group, the glucose area under the curve (AUC) for the OGTT in the DM+ST group exhibited a reduced glucose AUC by 14.7%. *p < 0.05.

According to the OGTT, although there were no significant differences before and 15, and 120 min after the glucose load between the CON group and CON+ST group, the blood glucose levels significantly decreased 30 and 60 min after the glucose load (p < 0.05, Fig. 3) in the CON+ST group as compared to those in the CON group. The CON+ST group exhibited a reduced glucose AUC by 18.2% (p < 0.05, Fig. 4) for the OGTT, compared with the CON group.

Figure 3 The effect of heat-killed S. thermophilus on blood glucose during the OGTT in the CON+ST and CON groups.

OGTT, oral glucose tolerance test. Error bars represent one standard deviation. According to the OGTT, although there were no significant differences before and 15, and 120 min after the glucose load between the CON group and CON+ST group, the blood glucose levels significantly decreased 30 and 60 min after the glucose load in the CON+ST group as compared to those in the CON group. *p < 0.05.

Figure 4 Area under the curve (AUC) for the OGTT in the CON+ST and CON groups.

The CON+ST group exhibited a reduced glucose AUC by 18.2% for the OGTT, compared with the CON group. *p < 0.05.

Fasting insulin, HbA1c, and HOMA-IR

The heat-killed S. thermophilus treatment reduced serum insulin levels, HbA1c, and HOMA-IR (p < 0.05, Table 1) in ZDF diabetic rats. However, compared to the CON group, the heat-killed S. thermophilus treatment rats failed to produce significantly lower serum insulin levels or HOMA-IR in the CON+ST group (p < 0.05, Table 2).

Serum biochemical parameters

There were no significant differences in the serum creatinine, alanine aminotransferase, carbamide, or uric acid levels between the two groups in ZDF diabetic rats. The level of TC significantly increased in the DM+ST group, while the heat-killed S. thermophilus treatment did not significantly reduce the TG, HDL-C, or LDL-C levels in diabetic rats (p < 0.05, Table 1). In contrast, there were no significant difference in serum biochemical parameters in the CON+ST group compared to the CON group (p > 0.05, Table 1).

Inflammatory factors

Compared with the DM group, the inflammatory factors LPS, IL-6, and TNF-α significantly decreased and IL-10 significantly increased in the DM+ST group (p < 0.05, Table 1). There were no significant differences in the inflammatory factors between the CON+ST group and CON group (p > 0.05, Table 2).

Histological analysis

We examined the heat-killed S. thermophilus effects on the villi length and crypt depth in the ileum. In the diabetic rats, the intestinal mucosal layer was characterized by disturbed mucosal architecture, shortened villi, blunted villus tips, and inflammatory cell infiltration. In the DM+ST group, oral administration of S. thermophilus restored the normal structure of the intestinal mucosal layer (Fig. 5). The length of villi and depth of crypts in the DM+ST group were significantly increased compared to those in DM group (Fig. 6). Additionally, goblet cells were counted per villus/crypt in the ileum. The ileum exhibited a significant increase in total goblet cell number after treatment with S. thermophilus (39.2 ± 4.2 vs 20.9 ± 5.0, p < 0.05). Similar findings were also seen in the colonic tissues (Figs. 5 and 7). There were no differences in the villi length and crypt depth and the numbers of goblet cells between the CON and CON+ST groups (Fig. 8).

Figure 5 Representative histology of the ileum and colon with HE stain in T2D model rats.

(A) Histology of the ileum in the DM group, (B) histology of the ileum in the DM+ST group, (C) histology of the colon in the DM group, and (D) histology of the colon in the DM+ST group. The image acquisition phase was performed with a 50× objective. Scale bar = 200 μm. In the diabetic rats, the intestinal mucosal layer was characterized by disturbed mucosal architecture, shortened villi, blunted villus tips, and inflammatory cell infiltration. In the DM+ST group, oral administration of S. thermophilus restored the normal structure of the intestinal mucosal layer.

Figure 6 The length of villi and depth of crypts in the ileum of diabetic rats.

The length of villi and depth of crypts in the ileum in the DM+ST group were significantly increased compared to those in DM group. *p < 0.05.

Figure 7 The length of villi and depth of crypts in the colon of diabetic rats.

The length of villi and depth of crypts in the colon in the DM+ST group were significantly increased compared to those in DM group. *p < 0.05.

Figure 8 Representative histology of the ileum and colon with HE stain in SD rats.

(A) Histology of ileum in the CON group, (B) histology of ileum in the CON+ST group, (C) histology of colon in the CON group, (D) histology of colon in the CON+ST group. The image acquisition phase was performed with a 50× objective. Scale bar = 200 μm. The characteristics of the intestinal mucosal layer were similar between the CON+ST group and CON group. There were no differences in the villi length and crypt depth and the numbers of goblet cells between the CON+ST group and CON group.

Western blot analysis

To explore the mechanisms underlying the heat-killed S. thermophilus effects on the barrier function, the expression levels of Occludin and ZO-1 proteins were determined by Western blot analysis. The results showed that Occludin and ZO-1 proteins in the DM+ST group were significantly elevated compared with the DM group both in the ileum and colon tissues (1.76-fold increases for Occludin and 2.29-fold increases for ZO-1 in the ileum tissues; 1.64-fold increases for Occludin and 1.46-fold increases for ZO-1 in the colon tissues, vs the DM group) (Fig. 9). There were no differences in the expression levels of Occludin and ZO-1 in the ileum or colon tissues between the CON and CON+ST groups (Fig. 10).

Figure 9 Effects of the heat-killed S. thermophilus treatment on tight junction proteins in the DM+ST and DM groups.

(A) Ileum and colon extracts from DM+ST and DM groups were used for Western blot analysis; (B) Expression levels of Occludin were quantified by measuring band densities; (C) Expression levels of ZO-1 were quantified by measuring band densities. β-actin was used as a loading control. *p < 0.05.

Figure 10 Effects of the heat-killed S. thermophilus treatment on tight junction proteins in the CON+ST and CON groups.

(A) Ileum and colon extracts from CON+ST and CON groups were used for Western blot analysis; (B) expression levels of Occludin were quantified by measuring band densities; (C) expression levels of ZO-1 were quantified by measuring band densities. β-actin was used as a loading control.

Characterization of gut microbiota

In ZDF diabetic rats, the richness of the gut microbiota was increased in the DM+ST group compared with the DM group; however, the difference was not significant, as shown in Table 3. Significant difference did not exist between the CON+ST group and the CON group. As shown in Fig. 11, to assess the bacterial community between two groups, a PCoA for the unweighted UniFrac distance matrices was performed. The first two principal coordinates of PCoA (components 1 and 2) were separated into DM+ST and DM groups, which shared overlapping regions. As in the above analysis, the DM+ST and DM groups, and COM+ST and CON groups exhibited similar alpha and beta diversities in the gut microbiota. The results indicate that the treatment with heat-killed S. thermophilus could not improve the richness of the gut microbiota.

Table 3 Alpha diversity indices.

	CON	CON+ST	DM	DM+ST	
OTUs	1,497.40 ± 327.41	1,504.20 ± 275.38	1,064.60 ± 230.90	1,298.60 ± 323.53	
Chao1	4,143.66 ± 490.57	4,218.70 ± 524.60	3,253.11 ± 518.16	3,185.49 ± 733.41	
Shannon	117.34 ± 11.09	121.01 ± 11.65	86.96 ± 8.13	90.94 ± 20.40	
Simpson	6.76 ± 0.66	6.88 ± 0.39	4.92 ± 0.70	5.72 ± 1.22	
PD_whole_tree	0.93 ± 0.04	0.93 ± 0.02	0.81 ± 0.08	0.87 ± 0.09	
Note:

Data are presented as means ± SD (n = 5). In ZDF diabetic rats, the richness of the gut microbiota was increased in the DM+ST group compared with the DM group; however, the difference was not significant (p > 0.05, t-test and Wilcoxon rank-sum test). Significant difference did not exist between the CON+ST group and the CON group (p > 0.05, t-test and Wilcoxon rank-sum test).

Figure 11 PCoA of unweighted UniFrac distances of the gut bacterial communities between the DM+ST and DM groups.

The first two principal coordinates of PCoA were separated into DM+ST and DM groups, which shared overlapping regions.

At the genus level, the abundance of Ruminococcaceae, Veillonella, Coprococcus, and Bamesiella was significantly elevated by heat-killed S. thermophilus treatment in ZDF diabetic rats (p < 0.05, Fig. 12), whereas Phascolarctobacterium and Dorea abundances were reduced by heat-killed S. thermophilus treatment in SD control rats (p < 0.05, Fig. 13).

Figure 12 The t-test results of the relative abundance (%) of bacteria from the DM+ST and DM groups.

At the genus level, the abundance of Ruminococcaceae, Veillonella, Coprococcus, and Bamesiella was significantly elevated by heat-killed S. thermophilus treatment in ZDF diabetic rats.

Figure 13 The t-test results of the relative abundance (%) of bacteria from the CON+ST and CON groups.

Compared with the CON group, Phascolarctobacterium and Dorea abundances were reduced by heat-killed S. thermophilus treatment in the CON+ST group.

Discussion

In this study, heat-killed S. thermophilus bacteria were administered to ZDF T2D rats to test whether they have a protective effect. The ZDF diabetic rat is a well-characterized model of T2D, and the rats has been used in many studies to examine human T2D pathophysiology and the effects of therapeutic options (Ferreira et al., 2010; Tikellis et al., 2004). Interestingly, we found that the heat-killed S. thermophilus treatment effectively moderated insulin resistance and glucose intolerance in the ZDF T2D rat model. To our knowledge, this is the first report about the effect of the heat-killed S. thermophilus treatment on glycemic parameters of diabetic rats. Many previous studies focused on the relation between live S. thermophilus and human. For instance, a multispecies probiotic supplement consisting of S. thermophilus reduced the fasting plasma glucose and serum high-sensitivity C-reactive protein, and increased plasma total glutathione (Asemi et al., 2013b). Also, the probiotic mix VSL#3, which contains S. thermophilus, increased insulin sensitivity, and affected the composition of gut microbiota (Rajkumar et al., 2014). So our work provides new insights into the function of the heat-killed S. thermophilus.

Another effect of the heat-killed S. thermophilus treatment is that of significantly reducing the level of TC in the diabetic rats used in this study. The effect of removing cholesterol probably occurs by two mechanisms: binding cholesterol to the cell surface (Kimoto, Ohmomo & Okamoto, 2002; Liong & Shah, 2005) or deconjugating bile salts to prevent their recycling (Iyer et al., 2010; Kim et al., 2017).

In addition, we found that the heat-killed S. thermophilus treatment increased the abundance of Ruminococcaceae, Veillonella, Coprococcus, and Barnesiella at the genus level in diabetic rats. The normal gut microbiota has many functions, such as protection against pathogens, immunomodulation, maintenance of the gut mucosal barrier structural integrity, and nutrient and drug metabolism (Jandhyala et al., 2015). As a member of short chain fatty acid producers, Ruminococcaceae is inversely correlated with increased intestinal permeability (Leclercq et al., 2014), and alcoholic cirrhosis (Bajaj et al., 2014). The abundance of Ruminococcaceae has been observed to significantly increase after treatment with fucoidan (Shang et al., 2016). Coprococcus is a butyrate-producing genera (Fujio-Vejar et al., 2017). Dietary intervention including extensively hydrolyzed casein formula supplemented with Lactobacillus rhamnosus GG to enrich Coprococcus could accelerate tolerance acquisition in infants who are allergic to milk (Berni et al., 2015). Veillonella are normal bacteria found in the intestines of mammals, that are well known for their lactate fermenting abilities. A positive association has been found between lactose levels and the abundance of the Veillonella genus (Pimentel et al., 2017). Anaerobic bacteria belonging to the Barnesiella genus enable clearance of intestinal colonization by the highly antibiotic-resistant bacterium vancomycin-resistant Enterococcus (Ubeda et al., 2013). When compared with high-fat, high-sucrose-fed mice, Barnesiella spp. are the main discriminative feature of chow-fed mice (Anhê et al., 2017). Therefore, Barnesiella may have a beneficial impact on host metabolism.

Many effects of probiotics are mediated through immune regulation and through the balance of anti-inflammatory and pro-inflammatory cytokines. In this study, the heat-killed S. thermophilus treatment significantly decreased the inflammatory factors LPS, IL-6, and TNF-α, and increased IL-10. From the membranes of gram-negative bacteria, LPS penetrates into the blood via impaired permeability of the intestinal mucosa, which is caused by the reduced expression of adhesion and tight junction proteins (Cani et al., 2008). Then, LPS triggers a strong pro-inflammatory reaction and secretion of proinflammatory cytokines from the host cells, followed by metabolic endotoxemia (Bäckhed et al., 2003). Metabolic endotoxemia increases systemic inflammation and impairs insulin sensitivity in both adipose tissue and the liver (Cani et al., 2007). It can also impair insulin signaling by inducing endoplasmic reticulum stress and the activity of a histone acetyltransferase (Cao et al., 2017). The high circulating LPS characterizes both incident and prevalent diabetes in a clinical observation also suggests the relevance of this putative mechanism to humans (Pussinen et al., 2011). As the product of pro-inflammatory cells, IL-6 is involved in many biological processes, such as the host response to acute-phase reactions, hematopoiesis, enteric pathogens, and terminal differentiation of B-lymphocytes (Adams, 2010). IL-10 is a potent deactivator of macrophage/monocyte proinflammatory cytokine synthesis (Clarke et al., 2015), such as downregulation of TNF-α secretion by macrophages (Fiorentino et al., 1991).

It was also found that the heat-killed S. thermophilus treatment protected the intestinal barrier. In our study, an increased ileum villus/crypt length and number of goblet cells were observed in the DM+ST group with S. thermophilus administration compared with the DM group. This is consistent with previous studies reporting that probiotic administration markedly deepened jejunal crypts in healthy rats (Tazuke et al., 2011), and both villus and crypt were lengthened after treatment by emu oil (Abimosleh et al., 2012). The main role of goblet cells is to protect the mucous membrane by secreting mucus (Robbe-Masselot et al., 2004). There is a strong association between intestinal flora and secretion of mucin (Yeung et al., 2015), as goblet cells may be regulated by interactions between the gastrointestinal mucosa and specific bacterial peptides (Leiper et al., 2001). The results of our study also showed that the Occludin and ZO-1 proteins in the DM+ST group were significantly elevated compared with the DM group both in the ileum and colon tissues. Intestinal barrier integrity is maintained by the tight junctions those are made of transmembrane, scaffold and adaptor proteins. Occludin is transmembrane protein embedded in the intracellular actin through attachment to adaptor protein ZO-1 (Bauer et al., 2010). It is widely reported that commensal bacteria have profound effects on epithelial integrity and permeability, particularly, on tight junctions maintenance (Alam & Neish, 2018). A dysbiosis adversely enhances intestinal permeability by modulating the expression of epithelial l tight junction proteins ZO-1 and Occludin (Cani et al., 2009). The mucosal barrier is very important for protecting the host tissue from damage that is mediated by toxic products or luminal pathogens obtained from food or pathogenic bacteria, while allowing uptake of nutrients at the same time. A previous study showed that feeding fermented milk produced by S. thermophilus and Bifidobacterium breve could reinforce the intestinal barrier (Terpend et al., 1999). Another study also showed that live S. thermophilus significantly increased the transepithelial electrical resistance in the intestinal Caco-2 cell monolayer by enhancement (actinin, occludin) or maintenance (actin, ZO-1) of cytoskeletal and tight junctional protein phosphorylation (Resta-Lenert & Barrett, 2003).

In the current study, the S. thermophilus used was heat-killed instead of live cells. Both live and dead cells are capable of generating a biological response (Dotan & Rachmilewitz, 2005). Our result is consistent with a recent study which shows that pasteurized Akkermansia muciniphila is able to ameliorate high-fat diet induced dysglycemia (Plovier et al., 2017). In a meta-analysis, modified (heat-killed or sonicated) probiotics were found to have effects similar to those of the living probiotics in most trials (Zorzela et al., 2017). The effects of heat-killed probiotics may be attributed to the dead cells and/or their metabolites. For example, metabolites released by S. thermophilus exerted an anti-TNF-α effect and were capable of crossing the intestinal barrier (Ménard et al., 2004). Besides, a recombinant protein isolated from the A. muciniphila membrane can lead to an improved gut barrier (Plovier et al., 2017). Notably, even A. muciniphila-derived extracellular vesicles can decrease gut permeability by regulating the tight junctions (Chelakkot et al., 2018). It has also been documented that bacterial muramyl dipeptide reduces inflammation and promotes insulin signaling in the state of metabolic endotoxemia, and glycemia (Cavallari et al., 2017). As a bacterial metabolite, indole is able to counteract the pro-inflammatory and metabolism-altering effects of LPS in the liver (Beaumont et al., 2018). Similarly, SCFAs can improve barrier function (Elamin et al., 2013), decrease inflammation, and promote the metabolism of lipids and glucose (Sonnenburg & Bäckhed, 2016; Canfora, Jocken & Blaak, 2015). In addition, microbiota-derived succinate can also improve glucose metabolism by acting on intestinal gluconeogenesis (De Vadder et al., 2016).

One limitation to widespread use of probiotic therapy is the concern regarding adverse effects, which may cause some pathology of their own (Berger, 2005). Compared with live probiotics, heat-killed probiotics are safer for purposes such as application in immunosuppressed patients and children (Vintiñi & Medina, 2011). Another problem with live probiotics is that they would have to survive proteolytic enzymes and the low pH of stomach acid. The recovery rate of total S. thermophilus from the terminal ileum of minipigs was very low after digesting a certain amount of live cells (Lick, Drescher & Heller, 2001). The preparation and administration of heat-killed probiotics are convenient compared to live probiotics (Josef & Ricardo, 2005). Products based on dead cells are easier to standardize, and store, and they also have a long shelf-life (Adams, 2010). Therefore, heat-killed probiotics may be a promising and safer alternative to live probiotics.

After the analysis of numerous studies, it was proposed that there may be a bacteria mucosal immunity-inflammation-diabetes (BMID) axis, through which herbal monomers and formulae improve diabetes (Gao et al., 2017). In this study, heat-killed S. thermophilus may also affect diabetes through the BMID axis by increasing the abundance of beneficial bacteria, protecting the intestinal epithelial barrier, and suppressing IL-6, LPS, and TNF-α secretion, and the end result is moderation of insulin tolerance.

Conclusion

Our study supports the hypothesis that treatment with heat-killed S. thermophilus could effectively improve the glycemic parameters of T2D model rats. In addition, the potential mechanisms underlying the protection may consist of changing the composition of gut microbiota, reinforcing the intestinal epithelial barrier and the immunity of the intestinal mucosa, decreasing the level of inflammation, and then reducing insulin resistance.

Supplemental Information

Supplemental Information 1 Raw data.

Raw data applied for data analyses and preparation for tables and figures.

Click here for additional data file.

Supplemental Information 2 Uncropped, unprocessed images of blots and gels of figure 9.

Click here for additional data file.

Supplemental Information 3 Uncropped, unprocessed images of blots and gels of figure 10.

Click here for additional data file.

We are very grateful to CapitalBio Technology Co., Ltd. for excellent technical assistance with 16s sequencing experiments, and thank LetPub for its linguistic assistance during the preparation of this manuscript.

Additional Information and Declarations

Competing Interests

Author Contributions

Animal Ethics

Data Availability

The authors declare that they have no competing interests.

Xiangyang Gao conceived and designed the experiments, performed the experiments, analyzed the data, prepared figures and/or tables, authored or reviewed drafts of the paper, approved the final draft.

Fei Wang conceived and designed the experiments, performed the experiments, prepared figures and/or tables, authored or reviewed drafts of the paper, approved the final draft.

Peng Zhao contributed reagents/materials/analysis tools, approved the final draft.

Rong Zhang contributed reagents/materials/analysis tools, approved the final draft.

Qiang Zeng conceived and designed the experiments, analyzed the data, prepared figures and/or tables, authored or reviewed drafts of the paper, approved the final draft.

The following information was supplied relating to ethical approvals (i.e., approving body and any reference numbers):

All experimental protocols were approved by the Animal Care Committee of the General PLA Hospital Animal Ethics Committee (project: CPLAGHAE-20171228-01).

The following information was supplied regarding data availability:

The raw data is available in the Supplemental Files and at Zenodo: Xiangyang Gao. (2019). Effect of heat-killed Streptococcus thermophilus on type 2 diabetes rats [Data set]. Zenodo. DOI 10.5281/zenodo.2553135.

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
