# Peer review of "Effect of heat-killed Streptococcus thermophilus on type 2 diabetes rats"

_PeerJ, doi:10.7717/peerj.7117_

## Round 0.1 · original submission · Minor Revisions

The manuscript addresses an important topic using straight-forward methods. However, a few points suggested by the reviewers should be addressed, in particular in the introduction and discussion, while English should be rechecked by a native English speaker.

Reviewer 1 ·

Basic reporting

Gao and colleagues present interesting data regarding the effect of an heat-inactivated extract of S. thermophilus on OGTT and low grade inflammation in diabetic rats.

Overall, the manuscript is nicely structured with an appropriate design according to the scientific question posed at the beginning.

The only weakness in this case are:
1- It is often mentioned that the effect of S. thermophilus on metabolism in humans is unknown. However, the study is on rats, thus I suggest to restructure this sentence removing the word human.

2- More in general, many sentences lack the preposition or, when used, it is wrong, see e.g. lines 88, 194, 196 and so on.. I suggest to involve an English native speaker to restructure sentences in a more appropriate fashion. Alternately, you can take advantage of English services available online.

3- Some more background information are needed in both the introduction section and in the discussion section. For instance, it is not introduced the role of the microbiota in promoting systemic and tissutal low grade inflammation through endotexemia, thus determining insulin sensitivity in liver and adipose tissue (Prattichizzo et al., 2018 Ageing Research Reviews x 2). Also, in the discussion section, some space is spent to describe previous work with S.thermophilus but not related to metabolism (e.g. intestinal infection). I suggest to remove these parts and rather comment relevant results obtained with different bacterial specied in similar settings. For instance, a recent paper (Plovier et al., 2017 Nat Med ) suggest that pasteurized Akkermansia Municiphila is able to ameliorate HFD induced dysglycemia, a finding similar to those of this paper . Your results are compatible with an effect mediated by a proteic product stimulating the epithelial barrier function of the colon, considering the lower circulating levels of LPS and of pro-inflammatory cytokines. The discussion could be more focused on this point.

Experimental design

The experimental design is ok to address the research question posed at the beginning. The methods are sufficiently detailed to guarantee replication. Ethical approval has been obtained.

Validity of the findings

Data appear robust and statistically appropriate. Graphs with individual points are preferable respect to histograms, but this is not mandatory. Conclusions are not overstated but the discussion cold be better balanced (see above).

The only experimental gap regard the histological analysis of the villi in the ileum and colon. These data should be further substantiated by expression data (mRNA expression through RT PCR or direct staining of proteins in tissue sections of the same tissues) regarding the expression levels of junction proteins, e.g. occludin, ZO-1 etc (see Plovier et al., 2017 Nat Med). These would be of utmost importance to hypothesize an effect on barrier function and endotexemia.

Reviewer 2 ·

Basic reporting

In this paper the Authors aim to evaluate the effect of heat-killed S. thermophilus on metabolic parameters using a diabetic rat model.
Below are listed some of my comments and suggestions:
- Figures numbering from Fig.3 onward does not match the order of their presentation in the text; (in particular, line 201 it is reported that fig 3 represents blood glucose levels after glucose load in CONT+ST group compared to CON group. However, this graph is missing).
- The discussion section should include a paragraph discussing the results on the effects of heat-killed S. thermophilus on blood glucose during a OGTT.
- Based on which information the dose of heat-killed S. thermophilus has been chosen? Please, indicate how long heat-killed S thermophilus has been administered to the rats.

Experimental design

The experiments are well designed; however, I only wonder if it would have been better to include an additional group treated with live microorganism. To the best of my knowledge, there are no studies investigating the effect of live S. thermophilus on glycemic parameters; and the Authors do not mention any reference in this regard.

Validity of the findings

The purpose of the work is interesting and still up to date. Researches aimed to shed light on the effect of “killed” probiotics and related-metabolites on human health and researches focused to indagate if live probiotics possess sufficient stability to survive the acid environment of the stomach and reach the intestine in intact form are of fundamental importance to understand the usefulness of probiotics preparations. In this regard, the study fits well into the scientific debate by providing evidence on the effectiveness of heat killed S. thermophilus on glycemic parameters.

---

## Round 0.2 · Major Revisions

The authors are invited to try to address the remaining issues raised by Reviewer 1, who comments that much of their prior feedback has not been addressed.

Reviewer 1 ·

Basic reporting

English is improved, as has the space dedicated to the topic of the paper. However, some citations could have been better choose, for example ref. n 10 reports the effect of dysbiosis on hepatic fat accumulation, rather than the development of insulin resistance. I rather suggest to use this other reference, since it is more appropriate to explain the proposed mechanism (doi: 10.1016/j.arr.2018.10.003).

Experimental design

Still no data showing the expression of junction protein in the intestine has been provided. So my comment remains the same of the first submission.

Validity of the findings

improved

Additional comments

Language is greatly improved. However, none of the other comments of this reviewer has been addressed. Still expression data of junction proteins are lacking and some relevant references are not cited. Thus, i suggest to perform another round of minor revision.

Reviewer 2 ·

Basic reporting

I have no further comments

Experimental design

I have no further comments

Validity of the findings

I have no further comments

Additional comments

I have no further comments

---

## Round 0.3 · accepted · Accept

The authors have carefully addressed all the remaining issues addressed by reviewer 1.

Reviewer 1 ·

Basic reporting

Ok

Experimental design

Ok

Validity of the findings

Ok

Additional comments

The authors satisfactorily addressed the comments of this reviewer. I suggest to accept the paper in this revised form.